# Extracted Plasma Cell-Free DNA Concentrations Are Elevated in Colic Patients with Systemic Inflammation

**DOI:** 10.3390/vetsci11090427

**Published:** 2024-09-12

**Authors:** Rosemary L. Bayless, Bethanie L. Cooper, M. Katie Sheats

**Affiliations:** 1Department of Molecular Biomedical Sciences, College of Veterinary Medicine, North Carolina State University, Raleigh, NC 27606, USA; rlbayles@ncsu.edu; 2Department of Clinical Sciences, College of Veterinary Medicine, North Carolina State University, Raleigh, NC 27606, USA; bplewis2@ncsu.edu; 3Comparative Medicine Institute, North Carolina State University, Raleigh, NC 27607, USA

**Keywords:** equine, systemic inflammatory response syndrome (SIRS), biomarker, gastrointestinal disease

## Abstract

**Simple Summary:**

Colic is a common and potentially life-threatening gastrointestinal (GI) condition in horses. Colic cases can be challenging for veterinarians to determine the cause, recommend appropriate treatment, and predict outcome. Biomarkers are molecules that can be measured in body fluids and can assist clinicians with disease diagnosis and/or prognosis. In this study, we investigated a biomarker known as cell-free DNA (cfDNA). This biomarker is released from cells when they are damaged, dying, or activated by inflammation. For this study, we measured the cfDNA in blood collected from horses with colic and compared cfDNA concentrations in groups of horses with different types of GI disease, different signs of inflammatory response, and survivors vs. non-survivors. We found that cfDNA was not different in comparisons of GI disease type or survival status, but cfDNA was elevated in colic patients that had evidence of systemic inflammation. Further research is needed to determine whether cfDNA is a clinically useful biomarker for colic in horses.

**Abstract:**

Colic is a common and potentially life-threatening condition in horses; in many cases, it remains challenging for clinicians to determine the cause, appropriate treatment, and prognosis. One approach that could improve patient care and outcomes is identification of novel diagnostic and prognostic biomarkers. Plasma cell-free DNA (cfDNA) is a biomarker that shows promise for characterizing disease severity and predicting survival in humans with acute abdominal pain or requiring emergency abdominal surgery. In horses, we recently determined that extracted plasma cfDNA concentrations are elevated in colic patients compared to healthy controls. For this current study, we hypothesized that extracted plasma cfDNA concentrations would be significantly higher in horses with strangulating or inflammatory colic lesions, in colic patients with systemic inflammatory response syndrome (SIRS), and in non-survivors. Cell-free DNA concentrations were measured in extracted plasma samples using a compact, portable Qubit fluorometer. Colic patients that met published criteria for equine SIRS had significantly higher median extracted plasma cfDNA compared to non-SIRS colic patients. There were no significant differences in extracted plasma cfDNA concentrations between other groups of interest. Our data offer early evidence that extracted plasma cfDNA concentration may provide information about systemic inflammation in colic patients, and additional research is warranted to expand on these findings.

## 1. Introduction

As the leading cause of mortality in adult horses, colic poses a considerable threat to equine well-being and survival [1]. Horses with strangulating intestinal lesions, and inflammatory conditions such as enteritis and colitis, experience particularly high rates of morbidity and mortality [2,3,4]. Challenging aspects of these cases include obtaining an accurate early diagnosis, advising owners on prognosis, making decisions regarding the need for intensive therapy and/or surgical intervention, and assessing a patient’s risk for complications. To address these challenges, numerous research studies have been conducted over the years to develop tools or models to aid in clinical decision-making for colic cases [5,6,7,8,9]. For example, one recent study of 451 colic cases found that packed cell volume at arrival, total plasma protein after surgery, and body condition score had the highest predictive power of case outcome [10]. Another area of research to aid colic diagnosis and prognosis is discovery and validation of biomarkers [11]. 

Investigation of biomarkers for colic severity and outcome has recently increased, and candidates for colic biomarkers in peripheral circulation include L-lactate, serum amyloid A (SAA), fibrinogen, creatine kinase, haptoglobin, procalcitonin, interleukin-6, and interleukin-1β [12,13,14,15,16,17,18,19,20,21,22,23]. Analytes that can be rapidly measured using point-of-care devices, such as lactate and SAA, are particularly desirable. Blood lactate at admission was higher in colic patients with strangulating lesions and in non-survivors [13,18,24]. However, other studies reported no association between admission blood lactate and presence of a strangulating lesion or likelihood of survival [12,25]. Elevation in peripheral SAA has been documented in colic patients with inflammatory lesions compared to those with non-inflammatory lesions [26,27] and in colic patients with clinical signs of systemic inflammatory response syndrome (SIRS) [28], but the evidence regarding SAA’s value for detecting intestinal strangulation and predicting outcome is conflicting [26,29]. Limitations of current colic biomarkers highlight the ongoing need for identification and investigation of novel biomarkers. One such potential biomarker is cell-free DNA (cfDNA). 

Plasma cfDNA originates from cells undergoing processes including apoptosis, necrosis, and neutrophil extracellular trap (NET) release [30]. In human medicine, plasma cfDNA has been widely studied as a minimally-invasive biomarker for detecting and characterizing neoplasia and fetal chromosomal abnormalities, monitoring organ transplant recipients and cancer patients during treatment, and predicting response to therapies, risk of complications, and mortality [31,32,33,34,35,36,37,38,39]. In human patients with abdominal pain [40] or requiring exploratory laparotomy [41], plasma cfDNA concentration at hospital admission was predictive of acute mesenteric ischemia, need for intensive care, and mortality. Cell-free DNA has also been investigated as a biomarker in veterinary species, and multiple studies report higher plasma cfDNA concentrations in dogs with sepsis, trauma, neoplasia, and immune-mediated hemolytic anemia compared to healthy controls [42,43,44,45]. One study determined that dogs with gastric dilatation-volvulus, a lesion with similar pathophysiological implications to large colon volvulus in horses, had significantly higher plasma cfDNA concentrations than healthy controls [46]. In horses, research on cfDNA has been limited. Fingerhut et al. (2019) documented significantly different median serum cfDNA concentration in horses with equine recurrent uveitis versus horses with healthy eyes [47]. Conversely, Colmer et al. (2021) found no difference in plasma cfDNA between septic, sick non-septic, and healthy neonatal foals [48]. In an equine model of equine carpal osteoarthritis, joints in which osteoarthritis was induced via arthroscopic creation of an osteochondral fragment had significantly higher median synovial fluid cfDNA compared to sham-operated joints at 4- and 9-weeks post-surgery [49]. Plasma cfDNA concentrations were not significantly different between osteoarthritis and control groups [49]. Recently, our group determined that extracted plasma cfDNA was elevated in colic patients compared to healthy horses [50]. This novel finding supports further investigation of cfDNA as a biomarker for equine colic. 

Until recently, the most common methods for investigation of circulating cfDNA required a laboratory environment for DNA extraction from plasma and specialized equipment such as a fluorescence plate reader or PCR systems for cfDNA quantification, which hindered the practicality of measuring cfDNA in clinical cases. But with the advent of portable, tabletop DNA measurement devices, such as the Qubit 4 Fluorometer (Invitrogen, Thermo Fisher Scientific, Waltham, MA, USA), analysis of individual patient samples is now feasible. The Qubit 4 Fluorometer is a compact instrument that offers simple and rapid (seconds per sample analysis) quantification of cfDNA. This instrument has been used to measure cfDNA directly in human and canine plasma without the need for extraction [43,51,52,53,54,55]. Qubit measurement of cfDNA directly in foal and adult horse plasma has also been reported [48,49], introducing the possibility of cfDNA as a point-of-care diagnostic tool in equine practice. However, our research has shown that direct measurement of cfDNA in adult horse or foal plasma with the Qubit device lacks accuracy, presumably due to interference from equine plasma matrix components [50,56]. This documented matrix effect may partially explain the lack of significant differences in plasma cfDNA between groups of interest in previous equine studies [48,49]. Therefore, similar to many human cfDNA studies [40,57,58], our investigations of Qubit measured cfDNA in adult equine plasma continue to require extraction as an initial step. 

The current study builds upon our laboratory’s previous work demonstrating increased extracted plasma cfDNA concentration in colic patients relative to healthy horses [50]. As the next step in evaluating the value of cfDNA in equine colic, we focused on clinically relevant groupings of colic patients. Our study objectives were to measure cfDNA in extracted plasma samples from equine colic patients using the Qubit 4 Fluorometer and to compare cfDNA concentrations between horses (1) diagnosed with strangulating, non-strangulating, or inflammatory colic lesions, (2) in SIRS (SIRS score ≥ 2) [59] versus non-SIRS (SIRS score 0 or 1) colic patients, and (3) in survivors vs. non-survivors. We hypothesized that extracted plasma cfDNA concentrations would be higher in horses with strangulating or inflammatory colic lesions versus non-strangulating lesions, SIRS colic patients versus non-SIRS patients, and non-survivors versus survivors. 

## 2. Materials and Methods

Animal use and study protocol were approved by the North Carolina State University Institutional Animal Care and Use Committee (Protocol 18-168-O); consent was obtained from all clients prior to horse enrollment. This was a prospective observational study. In our original *a priori* power calculations, we determined that we needed to enroll at least 10 individuals in each clinical group to detect a minimum difference in means of 10–20 ng/mL cfDNA, based on a one-way analysis of variance with a Type I error rate α = 0.05 and Power, 1 − β = 0.80, with an estimated standard deviation of 20 ng/mL. Based on similar criteria (20 ng/mL difference in means, SD 15 ng/mL), the minimum sample size for the Mann–Whitney U test was 10 per group.

### 2.1. Sample Collection and Processing

Admission blood samples were obtained from the jugular vein of adult horses (≥2 years old) presenting for acute colic to an academic veterinary referral hospital. Pregnant or post-partum (<1 month) mares and horses with known or suspected non-gastrointestinal sources of inflammation were excluded from the study. Blood was immediately placed into collection tubes containing K_3_EDTA plus a proprietary solution to prevent cell lysis and minimize ex vivo release of cfDNA (Cell-Free DNA Collection Tube; Roche Diagnostics GmbH, Mannheim, Germany). Tubes were refrigerated (4 °C) until separation of plasma by centrifugation at 2300× *g* for 10 min at room temperature as previously described. Plasma was harvested taking care not to disturb the buffy coat, and 1 mL aliquots were frozen at −80 °C until extraction. Sample processing and storage conditions were based on human and canine studies [60,61,62,63] and have previously been published by our laboratory [50].

### 2.2. DNA Extraction

DNA was extracted from 250 μL thawed plasma using a commercial kit (DNeasy Blood and Tissue Kit; Qiagen, Germantown, MD, USA) following a modified [64] manufacturer protocol and a standard 20 μL elution volume. 

### 2.3. Cell-Free DNA Measurement

Immediately after DNA extraction, concentrations of DNA were measured using a compact benchtop fluorometer (Qubit 4; Invitrogen, Thermo Fisher Scientific, Waltham, MA, USA). Buffers and reagents used for preparation of standards and experimental samples were from kits (Qubit 1X dsDNA HS Assay Kit; Invitrogen, Thermo Fisher Scientific, Waltham, MA, USA) supplied by the fluorometer manufacturer. Sample analysis was performed in accordance with manufacturer directions and as previously reported for DNA extracted from human or equine plasma [50,65,66,67,68,69]. The concentration of DNA in each sample was automatically calculated by the fluorometer algorithm based on fluorescence and dilution ratio. The DNA concentration reported by the instrument was converted to cfDNA concentration in the extracted plasma sample using the following equation [50]:$${\left[\mathrm{cfDNA}\right]}_{\mathrm{plasma}}=\frac{{\left[\mathrm{cfDNA}\right]}_{\mathrm{extracted}\;\mathrm{sample}\;}\times \;\mathrm{elution}\;\mathrm{volume}}{\mathrm{volume}\;\mathrm{of}\;\mathrm{extracted}\;\mathrm{plasma}}$$

Plasma cfDNA concentrations calculated from extracted samples were also expressed relative to the segmented neutrophil count in EDTA whole blood as previously described [43]:$${\mathrm{cfDNA}/\mathrm{neutrophil}\;\mathrm{ratio}\;(\mathrm{ng}/10}^{6}\;\mathrm{neutrophil})=\frac{{\left[\mathrm{cfDNA}\right]}_{\mathrm{plasma}\;}\left(\mathrm{ng}/\mathrm{mL}\right)}{{\mathrm{segmented}\;\mathrm{neutrophil}\;\mathrm{count}\;(\times 10}^{6}/\mathrm{mL})}$$

### 2.4. Clinical Data Collection and Characterization

Signalment, physical examination, hematology, and biochemistry findings at admission, primary lesion (suspected or confirmed), and survival to hospital discharge were recorded for all colic patients. Gastrointestinal lesions were categorized into three groups, strangulating, inflammatory, or non-strangulating, based on surgical or postmortem findings or clinicopathologic features. Inflammatory lesions were classified based on diagnosis of the attending veterinarians and included criteria commonly used to support diagnosis of enteritis, colitis, or peritonitis (e.g., more than one of fever, abnormal WBC count, hyperfibrinogenemia, hypoalbuminemia, increased intestinal wall thickness on ultrasound, abnormal reflux, diarrhea). Horses in which medical management was successful and that were not diagnosed with inflammatory lesions were considered to have had non-strangulating lesions. The SIRS score at the time of sample collection was calculated as described by Roy et al., based upon the following criteria: tachycardia (heart rate >52 beats per minute), tachypnea (respiratory rate >20 breaths per minute), abnormal temperature (<37 °C [98.6 °F] or >38.5 °C [101.3 °F]), and/or abnormal peripheral white blood count (<5 or >12.5 × 10^9^ cells/L) [59]. Each criterion was assigned a value of 1 (present) or 0 (absent), with a maximum score of 4 possible if horses met all of the four criteria. Horses with a total SIRS score of 0 or 1 were classified as non-SIRS, and horses with a total SIRS score of 2, 3, or 4 were classified as SIRS [59]. Patients for which a SIRS score or lesion category was not readily assignable based on available information, including non-survivors in which surgery or postmortem exam was not performed, were excluded from the SIRS status and lesion category analyses, respectively. Horses were considered non-survivors if they died or were euthanized prior to hospital discharge.

### 2.5. Statistical Analyses

Data groupings were examined for outliers (>3 standard deviations from the mean), and outliers were removed prior to further analyses. Normality of cfDNA and cfDNA/neutrophil ratio data was assessed using Shapiro–Wilk tests. One-tailed Student’s *t*-test or Mann–Whitney test, as appropriate, were used to compare mean or median cfDNA concentrations between groups. All analyses described in this section were performed using GraphPad Prism v. 9.3.0. Significance was set at *p* ≤ 0.05 for all analyses.

## 3. Results

Plasma was obtained from 67 colic patients between June-December 2019; a summary of signalment, suspected or confirmed lesion category, SIRS status, and short-term outcome is available in Table 1. Several outliers were excluded from selected comparisons based on the criteria described above; this was performed independently for each population of interest and sample type (e.g., an extracted plasma cfDNA value may have been truncated from groupings of SIRS status but included in comparisons of colic lesion category). Extracted plasma cfDNA concentrations were not normally distributed in any groups, and extracted plasma cfDNA/neutrophil ratio data were not normally distributed except for the inflammatory colic lesion group. Logarithmic transformation of the data did not produce consistent normal distributions. Therefore, results are reported as median (range), and nonparametric statistical tests were performed. The median plasma extracted cfDNA concentration for a subset (36 horses) of these colic patients has been previously published in our initial investigation comparing circulating cfDNA in colic patients versus healthy horses [50]. 

Extracted plasma cfDNA concentration and extracted plasma cfDNA/neutrophil ratio results are available in Appendix A. There was no significant correlation between age and extracted plasma cfDNA concentration across all colic samples (Spearman’s ρ: 0.08739, *p* = 0.4820, *n* = 67), and extracted plasma cfDNA concentrations were not significantly different in samples collected from mare colic patients versus gelding colic patients. There were no significant differences in ages or sex distributions between compared groups (e.g., among non-strangulating vs. strangulating vs. inflammatory lesions, between SIRS vs. non-SIRS colic cases, and between survivors vs. non-survivors). There was no significant difference in median extracted plasma cfDNA concentrations between pairwise comparisons of colic patients with non-strangulating (*n* = 38), strangulating (*n* = 13), or inflammatory lesions (*n* = 12, *p* ≥ 0.10 for all comparisons; Figure 1). Twenty-one colic patients met the criteria for equine SIRS (score ≥ 2) [59], and median cfDNA concentration for this group was significantly higher than for the non-SIRS colic cases (score 0–1, *n* = 39) (*p* = 0.01, Figure 2). Median extracted plasma cfDNA concentration was not higher in non-survivors (*n* = 25) versus survivors (*n* = 39; *p* = 0.10; Figure 3). 

The median extracted plasma cfDNA/neutrophil ratio was significantly higher in patients with inflammatory gastrointestinal disease (*n* = 11) than in cases with non-strangulating causes of colic (*n* = 38) or in horses with strangulating lesions (*n* = 11) (*p* < 0.0001 and *p* = 0.03, respectively; Appendix A). The median extracted plasma cfDNA/neutrophil ratio was also higher in colic patients diagnosed with SIRS (*n* = 21) versus non-SIRS colic patients (*n* = 39, *p* = 0.01; Appendix A). The median extracted plasma cfDNA/neutrophil ratios did not differ between survivors (*n* = 40) and non-survivors (*n* = 23; *p* = 0.48; Appendix A).

## 4. Discussion

Evidence from other species supports the potential of plasma cfDNA as a diagnostic and prognostic biomarker in human and canine patients with abdominal disease. Colic pathology, including inflammation and intestinal necrosis, could be associated with extracellular release of DNA. Recently, we found that extracted plasma cfDNA concentrations were significantly higher in colic patients compared to healthy controls [50]. This study is the next step for investigating the potential value of plasma cfDNA as a biomarker in colic patients. We hypothesized that extracted plasma cfDNA would be higher in horses with strangulating or inflammatory colic lesions, in colic patients with evidence of SIRS, and in colic patients that did not survive to hospital discharge. 

We found that colic patients that met published criteria for equine SIRS [59] had significantly higher median extracted plasma cfDNA concentration than colic patients not diagnosed with SIRS. This finding is in agreement with data from humans, which show higher mean plasma cfDNA concentrations in acute abdominal pain patients with SIRS versus without SIRS [40]. Higher plasma cfDNA was also reported in dogs with SIRS compared to healthy controls [43]. Septic human and canine patients, in which a diagnosis of SIRS was accompanied by suspected or confirmed bacterial infection, also had higher plasma cfDNA concentrations than sick, non-septic populations or healthy controls [40,42,43,70]. In contrast, two neonatal foal studies from different universities did not find a correlation between cfDNA concentrations measured in plasma, and the neonatal SIRS score, and median plasma cfDNA concentrations were not significantly different between healthy and septic foals [48,56]. In one of these studies, Hobbs et al. identified and accounted for plasma matrix effect, and still found no significant difference between extracted plasma cfDNA in septic and non-septic foals [56]. 

Regarding lesion diagnosis, higher median extracted plasma cfDNA concentration in colic patients with strangulating lesions versus inflammatory or non-strangulating lesions approached but did not reach significance within this study population. Additionally, there was considerable overlap in the extracted plasma cfDNA concentrations from horses within each lesion category. The overlap in extracted plasma cfDNA concentrations between the strangulating, non-strangulating, inflammatory lesion groups may reflect the fact that horses with more severe types of colic, including intestinal strangulation or marked intestinal inflammation, may be more rapidly presented to a referral facility compared to a more stable colic patient with non-strangulating cause of colic. If the severity of clinical signs prompts earlier veterinary attention and/or decision for referral, the admission blood samples from some of the patients with strangulating or inflammatory colic lesions may reflect earlier stages of the disease process compared to horses with a longer duration of non-strangulating lesions. Additionally, it is currently unknown what the primary source(s) of plasma cfDNA is in these patients (i.e., extrusion of NETs, necrosis, and/or apoptosis) and to what degree concentration of cfDNA measured in samples from jugular venipuncture are affected by patient variables such as poor tissue perfusion or volume resuscitation. 

In this study, there was no significant difference in median extracted plasma cfDNA between colic survivors and non-survivors. A similar lack of difference in plasma cfDNA was reported in dogs with gastric dilatation-volvulus that survived and those that did not survive [46]. These veterinary findings contrast with the literature on human patients with acute abdominal pain or suspected acute mesenteric ischemia, which shows that plasma cfDNA concentrations were higher in patients who died within a month of presentation compared to survivors [40,41]. Importantly, plasma cfDNA in these human patients was an independent predictor of one-month mortality. One major factor that may have contributed to the discrepancy between our outcome findings and those described in humans with abdominal diseases is the option for euthanasia of veterinary patients. All of the non-survivors in our population were euthanized according to clients’ decisions. It is possible that some colic patients that were euthanized may have survived given more intensive therapy or different owner circumstances or perspective. 

Since extrusion of NETs is a source for increased circulating cfDNA, a reduction in blood neutrophils has the potential to limit extracellular release of DNA into plasma. As neutropenia is not uncommon in colic patients experiencing systemic inflammation, we elected to consider whether expressing cfDNA relative to the segmented neutrophil count at presentation would improve the ability to detect a difference between groups of interest. Analysis of extracted plasma cfDNA/neutrophil ratio results showed a significant difference between patients with inflammatory colic lesions and those with non-strangulating or strangulating colic lesions. Similar to the extracted plasma cfDNA findings, colic patients meeting SIRS criteria had significantly higher median extracted plasma cfDNA/neutrophil ratio compared to non-SIRS colic patients. There was no significant difference in median extracted plasma cfDNA/neutrophil ratio in colic patients that survived to discharge versus non-survivors, so accounting for the effect of neutropenia on circulating cfDNA did not reveal significant differences between colic survivors and non-survivors. In our population, the number of patients with neutropenia (segmented neutrophil count <2.5 × 10^6^/mL blood) at presentation was relatively small in both the survivor (three neutropenic patients at presentation out of 40 survivors) and non-survivor (four neutropenic patients at presentation out of 23 non-survivors) groups. The expression of extracted plasma cfDNA relative to blood neutrophil count may have a greater impact (i.e., increased ability to detect differences between survivors and non-survivors compared to absolute extracted plasma cfDNA concentrations) among groups of horses with higher incidence of neutropenia at presentation.

Limitations of this study include characteristics of the sampled population and factors affecting patient outcome. Colic patients were enrolled upon presentation to an academic veterinary referral hospital, with many horses initially evaluated and treated by their primary veterinarian prior to referral. Therefore, horses in which extracted plasma cfDNA was measured often represented cases refractory to routine medical management in the field. Our results may not reflect the overall population of horses examined for colic in an ambulatory setting. Study samples were obtained from horses at a single institution and do not reflect geographical variation in horse demographics or incidence of specific types of colic. Definitive confirmation of colic lesion was not available in all cases. For survivors in which surgery or post-mortem exam was not performed, categorization of lesion type as non-strangulating or inflammatory was based on available clinical data, including diagnostic findings and diagnosis of the attending clinician. Surviving patients with insufficient or conflicting information to differentiate non-strangulating or inflammatory lesions were excluded from colic lesion category analysis. Non-survivors that did not undergo surgery or post-mortem exam were excluded from colic lesion category analysis. Despite exceeding the number of horses per group determined by a priori sample size calculations, sample sizes for some of the groups of interest (e.g., horses with strangulating or inflammatory colic lesions) may have limited our ability to detect differences between clinically relevant groups. Since all non-survivors were the result of euthanasia, factors other than patient status and disease status may have influenced short-term outcome. While there was no evidence that euthanasia of enrolled horses was elected solely for financial reasons, we recognize that financial constraints can have a substantial impact on clients’ decision-making, including whether to pursue intensive medical therapy or surgery, that may ultimately impact patient outcome. Additionally, desire to avoid perceived suffering or logistic concerns may have prompted clients to pursue euthanasia in horses with potentially treatable colic.

Detailed information regarding the duration of colic prior to hospital admission and sample collection was not available for each case, and as mentioned earlier, timing of sample collection relative to onset of colic could plausibly affect cfDNA concentrations. Other historical information, such as exercise history and severity of colic signs prior to admission, was also not consistently available. This information may be relevant to measurement of circulating cfDNA concentrations, as published data generally suggest a possible effect of muscle damage (as might occur secondary to extreme exertion or trauma) on circulating cfDNA concentrations. Several human studies have documented increases in serum or plasma cfDNA concentrations in subjects performing high-exertion activities [71,72,73]. In a small cohort of working farm dogs, plasma cfDNA was significantly increased following strenuous exercise [74]. However, Devall et al. report that serum cfDNA concentration was not significantly higher in sled dogs after a 1600 km race compared to prerace cfDNA concentrations [75]. Within this study’s population, serum creatine kinase (CK) concentration at admission was available for 63 of the 67 total horses; 23 of the 63 horses had CK above the reference interval (>470 IU/L) for the laboratory in which biochemistry analyses were performed. There was no significant correlation between elevated CK concentrations (>470 IU/L) and extracted plasma cfDNA concentrations in the same patient. Therefore, it is unlikely that DNA release from cells due to muscle injury is a substantial contributor to the extracted plasma cfDNA concentrations measured in this cohort. 

This study was designed to investigate significant differences in systemic cfDNA in clinically relevant categories of equine colic patients. Our finding that extracted plasma cfDNA is elevated in colic patients that meet criteria for SIRS is novel. The ability to predict and/or detect systemic inflammation in early stages, possibly preceding abnormalities in traditional markers of SIRS (e.g., vital parameters, blood leukocyte count), could support informed decision-making and may improve patient outcomes and/or client satisfaction. Of particular interest for future studies is measurement of plasma cfDNA upon first colic evaluation by the primary care veterinarian. Identification of a biomarker with high sensitivity and/or specificity for colic severity that could be measured “on the farm” could have a significant impact on clinical practice and equine health. Although direct measurement of cfDNA in whole equine plasma with the Qubit is inaccurate, our lab is investigating dilution as a possible method to decrease autofluorescence and off-target fluorescence marker binding. We also plan to approach industry representatives to discuss optimizing a Qubit-style device for measurement of cfDNA in equine plasma. In other future studies, longitudinal measurement of extracted plasma cfDNA in horses with gastrointestinal disease would be useful to evaluate the kinetics of circulating cfDNA, especially as it relates to clinical status (e.g., changes in physical exam and hematological findings, development of SIRS-associated complications, patient outcome). This type of data would also be valuable for other disease processes associated with SIRS, including pneumonia/pleuropneumonia, retained fetal membranes, and neonatal foal sepsis. Our ultimate goal is to evaluate whether measurement of circulating cfDNA could improve existing diagnostic, prognostic, and/or predictive biomarkers/models for equine patients at risk of SIRS, especially when combined with other clinicopathological variables. 

Unfortunately, the documented matrix effect of equine plasma currently necessitates DNA extraction prior to cfDNA measurement, which currently precludes use of the Qubit as a “stall-side” diagnostic for plasma samples. Our lab is working to develop plasma preparation methods, such as dilution, that are feasible in a clinical setting. We are also investigating peritoneal fluid as an alternative sample type that more closely reflects intestinal pathology and allows rapid cfDNA quantification without an extraction step. Collaborative, multi-center studies will be an important strategy for a more comprehensive assessment of the value of plasma and peritoneal cfDNA as a biomarker in colic patients or horses with other systemic diseases.

## 5. Conclusions

This study investigated significant differences in systemic cfDNA in clinically relevant categories of equine colic patients. Our finding that extracted plasma cfDNA is elevated in colic patients that meet the criteria for SIRS is novel. The ability to predict and/or detect systemic inflammation in early stages, possibly preceding abnormalities in traditional markers of SIRS (e.g., vital parameters, blood leukocyte count), could support informed decision-making and may improve patient outcomes and/or client satisfaction.

## Figures and Tables

**Figure 1 vetsci-11-00427-f001:**
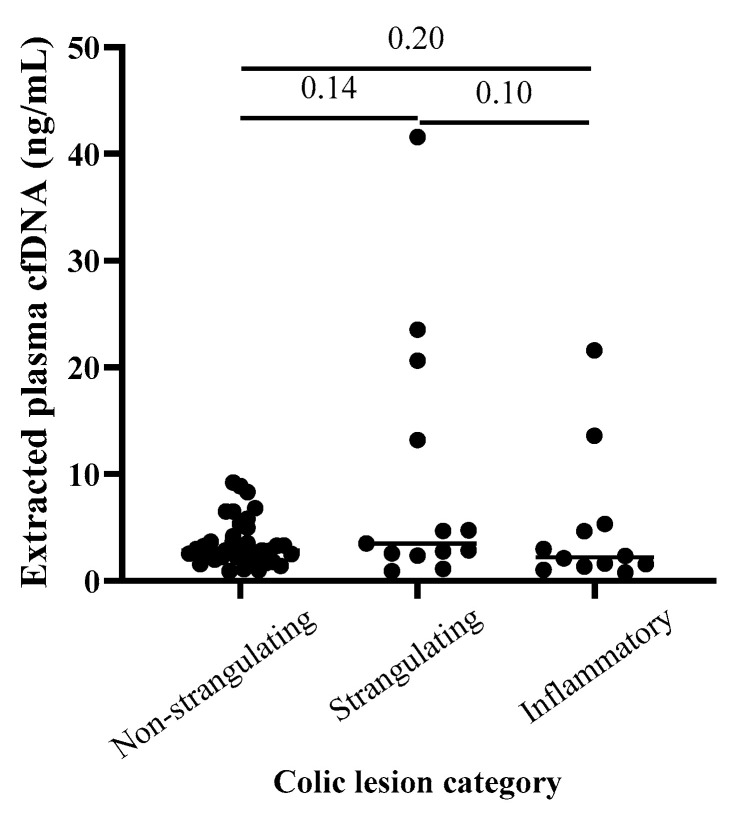
**Comparison of extracted plasma cell-free DNA (cfDNA) concentrations between colic lesion categories.** Differences in extracted plasma cfDNA between horses with non-strangulating (*n* = 38), strangulating (*n* = 13), or inflammatory (*n* = 12) colic lesions did not reach statistical significance. One-tailed Mann–Whitney test.

**Figure 2 vetsci-11-00427-f002:**
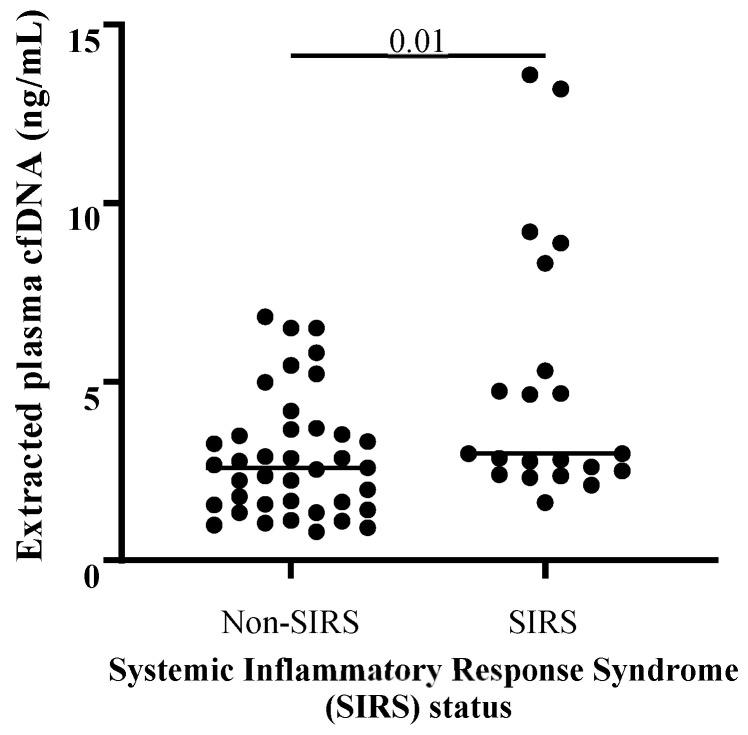
Comparison of extracted plasma cell-free DNA (cfDNA) concentrations between SIRS colic patients and non-SIRS colic patients. Horses that met the criteria for equine SIRS (*n* = 21) had significantly higher median extracted plasma cfDNA concentrations than colic patients without evidence of SIRS (*n* = 39). One-tailed Mann–Whitney test.

**Figure 3 vetsci-11-00427-f003:**
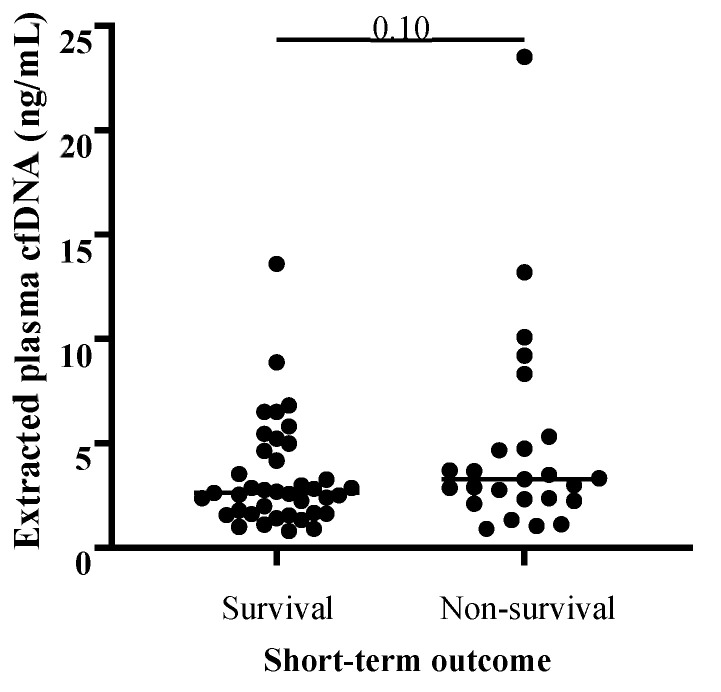
**Comparison of extracted plasma cell-free DNA (cfDNA) concentrations between colic non-survivors and survivors.** Median extracted plasma cfDNA concentrations were not significantly different between colic patients that did not survive to hospital discharge (*n* = 25) and colic survivors (*n* = 39). One-tailed Mann–Whitney test.

**Table 1 vetsci-11-00427-t001:** Signalment, suspected or confirmed lesion category, systemic inflammatory response syndrome (SIRS) status, and short-term outcome for 67 colic patients. * Numbers in each grouping may differ from sample sizes reported for individual comparisons in text, figures, and other tables due to exclusion of outlier data points for certain groupings. ^†^ Includes Appendix Quarter Horses (Quarter Horse/thoroughbred cross) and Paint breeds. ^#^ Lesion category or SIRS status not available for some patients due to insufficient information.

Variable	Colic Patients (*N* = 67) *
Age (median, range)	12 years (2–35 years)
Sex (number of horses)	28 Mare
	39 Gelding
	0 Stallion
Breed (number of horses)	14 Quarter Horse and Related Breed ^†^
	14 Warmblood
	13 Thoroughbred
	6 Arabian
	5 Draft Horse
	4 Pony
	2 Andalusian
	2 Friesian/Friesian Cross
	2 Paso Fino
	1 Miniature Horse
	1 Morgan
	1 Mule
	1 Norwegian Fjord
	1 Saddlebred
Lesion category	38 Non-strangulating
	13 Strangulating
	12 Inflammatory
	4 n/a ^#^
SIRS status	40 Non-SIRS
	22 SIRS
	5 n/a ^#^
Short-term outcome	41 Survival
(survival to discharge)	26 Non-survival

## Data Availability

The original contributions presented in the study are included in the article and Appendix A; further inquiries can be directed to the corresponding author.

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
