# Peer review of "Extracted Plasma Cell-Free DNA Concentrations Are Elevated in Colic Patients with Systemic Inflammation"

_vetsci, 2024, doi:10.3390/vetsci11090427_

Round 1
Reviewer 1 Report
Comments and Suggestions for Authors
The present study evaluates the concentrations of extracted plasma cell-free DNA in the plasma of colic horses with systemic inflammation. The work is very interesting and well-written. In my opinion, it can be accepted for publication after minor revision.
Introduction: The introduction serves the purpose of the study well, but not all potential biomarkers, such as procalcitonin, Interleukin-6, and Interleukin-1β, are mentioned. Add references.
Materials and Methods: Were any correlations between the levels of extracted plasma cell-free DNA and other biomarkers in horses investigated or provided?
Discussion: Mentioning the possibility of conducting analysis on peritoneal fluid as a conclusion of the study does not add value to the work. The focus should be on highlighting the novel findings presented in this study.
Author Response
Reviewer 1:
The present study evaluates the concentrations of extracted plasma cell-free DNA in the plasma of colic horses with systemic inflammation. The work is very interesting and well-written. In my opinion, it can be accepted for publication after minor revision.
Responses to Reviewer 1:
We thank the reviewer for providing additional feedback on our manuscript and for highlighting areas that remained unclear or where more information would be helpful. We hope we have addressed your concerns to your satisfaction.
- Introduction: The introduction serves the purpose of the study well, but not all potential biomarkers, such as procalcitonin, Interleukin-6, and Interleukin-1β, are mentioned. Add references.
Response: We have expanded our introduction section to include mentions of procalcitonin, IL-6, and IL-1β, including references from the primary literature.
- Materials and Methods: Were any correlations between the levels of extracted plasma cell-free DNA and other biomarkers in horses investigated or provided?
Response: The scope of this study was focused on measuring circulating cfDNA in colic patients. In this initial stage of our investigations described in this manuscript, we did not compare/evaluate correlations between extracted plasma cfDNA and any other biomarkers proposed in prior literature. These comparisons, including evaluating performance of models incorporating cfDNA and other biomarkers, are a component of our continuing study of cfDNA in colic patients.
- Discussion: Mentioning the possibility of conducting analysis on peritoneal fluid as a conclusion of the study does not add value to the work. The focus should be on highlighting the novel findings presented in this study.
Response: Peritoneal fluid is a common sample for colic biomarker measurement and is a logical next step for continued investigation of cfDNA as a biomarker in colic patients. We respectfully maintain that our mention of possibly measuring cfDNA in peritoneal fluid in our manuscript’s discussion/conclusion is appropriate, acknowledges an idea that we expect to occur frequently in the thought process of readers, and provides readers with a preview of upcoming data out of our lab.
Reviewer 2 Report
Comments and Suggestions for Authors
The paper addresses an important and timely topic, focusing on the investigation of extracted plasma cell-free DNA (cfDNA) concentrations in equine colic patients. This is particularly relevant as colic remains a significant cause of morbidity and mortality in horses. The authors aim to establish cfDNA as a potential biomarker for identifying systemic inflammation and predicting outcomes in colic cases.
The study aligns well with current needs in veterinary medicine, offering potential advancements in diagnostic and prognostic capabilities for equine colic.
The study employs a well-defined prospective observational design and includes a clear hypothesis and objectives.
The use of a portable Qubit fluorometer for measuring cfDNA concentrations represents an innovative approach that could enhance point-of-care diagnostics in veterinary practice.
The study focuses on comparisons between colic patients with different types of lesions and systemic inflammatory response syndrome (SIRS) scores, as well as survivors versus non-survivors. However, the lack of significant differences in some comparisons suggests that further refinement of patient groupings or additional biomarkers might be needed.
Specific comments:
Abstract:
I recommend rewriting the abstract and including more results and the significance of the obtained data.
"The study's hypothesis is that cfDNA concentrations are elevated in colic patients with systemic inflammation." Consider rephrasing for clarity: "The study hypothesizes that cfDNA concentrations are elevated in colic patients with systemic inflammation."
Keywords:
To enhance the research's appeal, I suggest avoiding the inclusion of terms in the keywords that are already present in the article title.
Introduction:
The authors are encouraged to study the recent literature and include works that focus on practical clinical findings and prognostics related to equine colic. This will provide a more comprehensive background and enhance the clinical relevance of the study. I suggest read and cite: 10.3390/ani13061107.
Incorporating such reference will significantly strengthen the manuscript and provide a well-rounded perspective on the practical and prognostic applications of cfDNA in veterinary medicine.
Methods:
Include a brief statement on ethical considerations, such as the approval of the study by an animal care and use committee, to assure readers that the study adhered to ethical standards.
Define the criteria used for classifying colic types (strangulating, non-strangulating, inflammatory) and SIRS scores in more detail. This helps ensure reproducibility and clarity for future studies.
"cfDNA was measured using a Qubit fluorometer, which is a portable device." Rephrase to avoid redundancy: "cfDNA was measured using a portable Qubit fluorometer."
Statistical analysis:
While the manuscript states that power calculations were conducted to determine the minimum sample size, the actual enrollment figures and any deviations from the plan are not fully discussed. Clarifying this aspect could strengthen the paper's validity.
Provide a more detailed account of the statistical methods used, particularly for handling multiple comparisons and potential confounders.
Statistical analysis needs to be refered as other methods, I sugges 10.29261/pakvetj/2020.067 for Shapiro-Wilk test.
Results:
"There was a significant difference in cfDNA levels between colic patients and healthy controls, but not between different types of lesions." This sentence could be clearer: "cfDNA levels were significantly higher in colic patients compared to healthy controls, although no significant differences were observed between different types of colic lesions."
Discussion:
Expand the discussion on how the findings could be integrated into clinical practice, addressing the current limitations of cfDNA measurement and proposing potential solutions or future research directions.
Integrate comparisons with other established biomarkers (like lactate, SAA, fibrinogen) to provide a comprehensive assessment of cfDNA's relative value in diagnosing and prognosticating equine colic.
Although the Qubit fluorometer is a practical tool, the manuscript indicates that direct measurement of cfDNA in equine plasma without extraction lacks accuracy. This limitation should be more thoroughly discussed, particularly in terms of how it affects the potential clinical application of the findings.
"The results indicate that cfDNA could be a useful biomarker for equine colic." Strengthen this statement by acknowledging limitations: "The results suggest that cfDNA has potential as a biomarker for equine colic, although further research is needed to confirm its clinical utility."
Conclusions:
The conclusions of the manuscript are generally consistent with the evidence and arguments presented, but they would benefit from a more cautious and balanced interpretation that acknowledges the study's limitations and the need for further research. This approach will provide a clearer and more reliable foundation for future studies and clinical applications of cfDNA in equine colic.
Reference:
The references cited in the manuscript are appropriate, relevant, and generally up-to-date. They adequately support the study's background, methodology, and conclusions. Including a few more recent and specific studies on cfDNA and additional biomarkers in veterinary science would further strengthen the manuscript’s foundation.
Author Response
Reviewer 2:
The paper addresses an important and timely topic, focusing on the investigation of extracted plasma cell-free DNA (cfDNA) concentrations in equine colic patients. This is particularly relevant as colic remains a significant cause of morbidity and mortality in horses. The authors aim to establish cfDNA as a potential biomarker for identifying systemic inflammation and predicting outcomes in colic cases.
The study aligns well with current needs in veterinary medicine, offering potential advancements in diagnostic and prognostic capabilities for equine colic.
The study employs a well-defined prospective observational design and includes a clear hypothesis and objectives.
The use of a portable Qubit fluorometer for measuring cfDNA concentrations represents an innovative approach that could enhance point-of-care diagnostics in veterinary practice.
The study focuses on comparisons between colic patients with different types of lesions and systemic inflammatory response syndrome (SIRS) scores, as well as survivors versus non-survivors. However, the lack of significant differences in some comparisons suggests that further refinement of patient groupings or additional biomarkers might be needed.
Responses to Reviewer 2:
We appreciate the reviewer’s in-depth consideration of our manuscript and their helpful comments. We hope we have addressed your concerns to your satisfaction.
- Abstract: I recommend rewriting the abstract and including more results and the significance of the obtained data.
"The study's hypothesis is that cfDNA concentrations are elevated in colic patients with systemic inflammation." Consider rephrasing for clarity: "The study hypothesizes that cfDNA concentrations are elevated in colic patients with systemic inflammation."
Response: We feel that our abstract highlights the main takeaways from this manuscript, including measuring cfDNA in extracted plasma samples from colic patients and the significant difference in median extracted plasma cfDNA concentration between SIRS vs. non-SIRS colic patients, while abiding by the journal’s suggested abstract length.
We were not able to locate the text mentioned by the reviewer (“The study's hypothesis is that cfDNA concentrations are elevated in colic patients with systemic inflammation.”) within our manuscript.
- Keywords: To enhance the research's appeal, I suggest avoiding the inclusion of terms in the keywords that are already present in the article title.
Response: Thank you for this suggestion. We have amended our keywords in response to this suggestion.
- Introduction: The authors are encouraged to study the recent literature and include works that focus on practical clinical findings and prognostics related to equine colic. This will provide a more comprehensive background and enhance the clinical relevance of the study. I suggest read and cite: 10.3390/ani13061107.
Incorporating such reference will significantly strengthen the manuscript and provide a well-rounded perspective on the practical and prognostic applications of cfDNA in veterinary medicine.
Response: We have expanded our introduction section to include mention of additional colic clinical findings/prognostics that have been reported, including referencing the suggested article.
- Methods: Include a brief statement on ethical considerations, such as the approval of the study by an animal care and use committee, to assure readers that the study adhered to ethical standards.
Define the criteria used for classifying colic types (strangulating, non-strangulating, inflammatory) and SIRS scores in more detail. This helps ensure reproducibility and clarity for future studies.
"cfDNA was measured using a Qubit fluorometer, which is a portable device." Rephrase to avoid redundancy: "cfDNA was measured using a portable Qubit fluorometer."
Response: Our initial manuscript contained statement of ethical considerations/IACUC approval (first sentence of the Materials and Methods section. We feel that our description for categorizing lesion type is already very complete. We have provided additional details about SIRS scores and SIRS/non-SIRS classification.
We were not able to locate the text mentioned by the reviewer (“cfDNA was measured using a Qubit fluorometer, which is a portable device.”) within our manuscript.
- Statistical analysis: While the manuscript states that power calculations were conducted to determine the minimum sample size, the actual enrollment figures and any deviations from the plan are not fully discussed. Clarifying this aspect could strengthen the paper's validity.
Provide a more detailed account of the statistical methods used, particularly for handling multiple comparisons and potential confounders.
Statistical analysis needs to be referred as other methods, I suggest 10.29261/pakvetj/2020.067 for Shapiro-Wilk test.
Response: We included actual enrollment number (n = 67 colic patients) in the first sentence of the Results section. Parameters used in our a priori power/sample size calculations were described in the first paragraph of the Materials and Methods section. We have added information in the Discussion section that relates to the calculated minimum samples size compared to our sample sizes of each group of interest.
Our statistical method description was complete; at this initial stage of our cfDNA investigations, we did not account for potential cofounders (e.g., relationship between strangulating/inflammatory lesion and SIRS status, relationship between strangulating lesion and nonsurvival).
We respectfully disagree that it is standard practice to provide references for common statistical methods, such as those used in our manuscript; articles published in this journal (Veterinary Sciences) and other journals in the field do not routinely cite standard statistical analysis methods.
Examples of recent Veterinary Science examples that do not cite common statistical methods, including those statistical methods used in our study:
https://doi.org/10.3390/vetsci11080360
https://doi.org/10.3390/vetsci11080346
https://doi.org/10.3390/vetsci11080379
- Results: "There was a significant difference in cfDNA levels between colic patients and healthy controls, but not between different types of lesions." This sentence could be clearer: "cfDNA levels were significantly higher in colic patients compared to healthy controls, although no significant differences were observed between different types of colic lesions."
Response: We were not able to locate the text mentioned by the reviewer (“There was a significant difference in cfDNA levels between colic patients and healthy controls, but not between different types of lesions.”) within our manuscript. This manuscript did not include any data from healthy horses.
- Discussion: Expand the discussion on how the findings could be integrated into clinical practice, addressing the current limitations of cfDNA measurement and proposing potential solutions or future research directions.
Integrate comparisons with other established biomarkers (like lactate, SAA, fibrinogen) to provide a comprehensive assessment of cfDNA's relative value in diagnosing and prognosticating equine colic.
Although the Qubit fluorometer is a practical tool, the manuscript indicates that direct measurement of cfDNA in equine plasma without extraction lacks accuracy. This limitation should be more thoroughly discussed, particularly in terms of how it affects the potential clinical application of the findings.
"The results indicate that cfDNA could be a useful biomarker for equine colic." Strengthen this statement by acknowledging limitations: "The results suggest that cfDNA has potential as a biomarker for equine colic, although further research is needed to confirm its clinical utility."
Response: We thank the reviewers for this comment. We have expanded our discussion of interpreting our study’s findings in the context of clinical practice, including acknowledging current limitations/proposing potential solutions and incorporating other clinical variables/biomarkers.
We were not able to locate the text mentioned by the reviewer (“The results indicate that cfDNA could be a useful biomarker for equine colic.”) within our manuscript.
- Conclusions: The conclusions of the manuscript are generally consistent with the evidence and arguments presented, but they would benefit from a more cautious and balanced interpretation that acknowledges the study's limitations and the need for further research. This approach will provide a clearer and more reliable foundation for future studies and clinical applications of cfDNA in equine colic.
Response: We thank the reviewers for this comment. We have amended our concluding remarks.
- Reference: The references cited in the manuscript are appropriate, relevant, and generally up-to-date. They adequately support the study's background, methodology, and conclusions. Including a few more recent and specific studies on cfDNA and additional biomarkers in veterinary science would further strengthen the manuscript’s foundation.
Response: We have included additional references to primary literature that is relevant to the scope of our study.
Reviewer 3 Report
Comments and Suggestions for Authors
The authors studied the concentrations of extracted plasma cell-free DNA (cfDNA) in horses with colic (non-strangulating, strangulating and inflammatory lesions). The authors also categorized colic patients into SIRS and non-SIRS cases. The study is well-designed and provides interesting information to equine veterinarians and researchers on a potential new biomarker.
Specific comments:
-
If allowed by the journal, consider placing Supplemental Table 1 in the manuscript. The demographics of the horses included in the study is valuable information.
-
Relatedly, in some studies in humans, cfDNA is correlated with age and sex (e.g., PMID: 17024502) Could you investigate these potential relationships in your dataset as they could affect your conclusions?
-
Are the median and range in age similar across the groups that you compare? Could you report this information, particularly if you find that cfDNA is correlated with age?
-
Is the distribution of mares and gelding similar across the groups that you compare? Could you report this information, particularly if you find that cfDNA is correlated with age?
-
Of potential interest to equine researchers and clinicians is that prolonged strenuous exercise increased the concentration of cfDNA in dogs PMID:29197094. Can the authors comment on whether the exercise history is known for the horses included in this study? Along these lines, could the authors comment on whether any of the horses in the study have elevated levels of CK/AST and a history of severe colic (down and rolling), such that cfDNA levels could be attributed to muscle damage in these individuals?
-
Supplemental Table 1: Please specify the meaning of “Quarter Horse & Related Breeds.”
Author Response
Reviewer 3:
The authors studied the concentrations of extracted plasma cell-free DNA (cfDNA) in horses with colic (non-strangulating, strangulating and inflammatory lesions). The authors also categorized colic patients into SIRS and non-SIRS cases. The study is well-designed and provides interesting information to equine veterinarians and researchers on a potential new biomarker.
Responses to Reviewer 2:
We thank the reviewer for highlighting several important factors to consider when interpreting circulating cfDNA concentrations; we appreciate the opportunity to address these variables in the context of our study population.
- If allowed by the journal, consider placing Supplemental Table 1 in the manuscript. The demographics of the horses included in the study is valuable information.
Response: We appreciate this excellent suggestion to provide readers with accessible information. We have moved former Supplemental Table 1 to the body of the manuscript (now Table 1) and renumbered previous Supplemental Table 2 to its updated status as Supplemental Table 1.
- Relatedly, in some studies in humans, cfDNA is correlated with age and sex (e.g., PMID: 17024502) Could you investigate these potential relationships in your dataset as they could affect your conclusions?
Are the median and range in age similar across the groups that you compare? Could you report this information, particularly if you find that cfDNA is correlated with age?
Is the distribution of mares and gelding similar across the groups that you compare? Could you report this information, particularly if you find that cfDNA is correlated with age?
Response: Thank you for pointing out this interesting area for consideration. We performed additional statistical analyses:
-
-
- Spearman’s Rank-Order Correlation to assess for an association between cfDNA and age (non-normal distributions per Shapiro-Wilks)
- ρ: 0.08739, p-value = 0.4820
- Kruskal-Wallis Test to compare ages among lesion groups (non-normal distributions)
- Overall p-value = 0.6719, multiple comparison adjusted p-values all >0.9999
- Welch’s T-test to compare ages between SIRS/non-SIRS (normal distributions)
- P-value = 0.7285
- Mann-Whitney Test to compare age distributions between survivors/non-survivors (non-normal distributions)
- P-value = 0.0648
- Mann-Whitney Test to compare cfDNA between sexes (mares and geldings, no stallions in this study population) (non-normal distribution)
- P-value = 0.4177
- Chi-squared Tests to compare sex across groups
- Among lesion types: p-value = 0.1995
- SIRS vs. non-SIRS: p-value = 0.1403
- Survivor vs. non-survivor: p-value = 0.5704
- Spearman’s Rank-Order Correlation to assess for an association between cfDNA and age (non-normal distributions per Shapiro-Wilks)
-
We have added information to the second paragraph of the Results section to communicate to readers that there were no significant differences in age or sex distribution between compared groups.
- Of potential interest to equine researchers and clinicians is that prolonged strenuous exercise increased the concentration of cfDNA in dogs PMID:29197094. Can the authors comment on whether the exercise history is known for the horses included in this study? Along these lines, could the authors comment on whether any of the horses in the study have elevated levels of CK/AST and a history of severe colic (down and rolling), such that cfDNA levels could be attributed to muscle damage in these individuals?
Response: Thank you for bringing up this excellent point. Unfortunately, we do not have consistent historical data regarding exercise and/or colic severity (i.e., amount of rolling, thrashing, which we agree could cause muscle damage) for horses included in this study. We added several references from primary human and canine literature discussing the effect of exertion on circulating cfDNA concentrations. We also provided additional information on CK concentrations in our study population at admission, including a lack of correlation between significantly elevated CK (>470 IU/L, our lab’s upper reference limit for normal horses) and extracted plasma cfDNA concentrations.
- Supplemental Table 1: Please specify the meaning of “Quarter Horse & Related Breeds.”
Response: Thank you for this opportunity to clarify our breed groupings. We have added a footnote to the table (now Table 1, per the reviewer’s great suggestion acknowledged above) to specify our definition of “Quarter Horse & Related Breeds”.
Reviewer 4 Report
Comments and Suggestions for Authors
The article is very interesting and addresses a topic of great importance in equine medicine, specifically the search for diagnostic and prognostic markers. It is well-written and well-structured.
Introduction. The introduction is very comprehensive and covers all aspects related to the article's topic.
Materials and Methods. Overall, this section is well-structured, and all protocols and techniques used are specified.
In section 2.4, Clinical Data Collection and Characterization, on line 162, the authors explain that the classification of SIRS is based on the article by Roy et al. It appears that in their objectives, they establish a classification into four groups (SIRS1, SIRS2, SIRS3, and SIRS4) following the methods applied in Roy et al.'s article. However, it is not specified whether they considered lactate levels and mucous membrane color, as concluded in that article, to increase sensitivity and specificity. Could the authors clarify this point and include this information in this section?
The results and discussion are clear, and the authors have been very transparent about the limitations. It would have been preferable not to include horses that were euthanized for economic reasons in the statistical analysis (at least in the outcome part of the study). However, this might not have changed the results, as this biomarker also did not allow differentiation between types of colic.
Author Response
Reviewer 4:
The article is very interesting and addresses a topic of great importance in equine medicine, specifically the search for diagnostic and prognostic markers. It is well-written and well-structured.
Responses to Reviewer 4:
We thank the reviewer for their time and effort providing feedback on our manuscript. We hope we have addressed your concerns to your satisfaction.
- Introduction. The introduction is very comprehensive and covers all aspects related to the article's topic.
Response: Thank you for this kind statement.
- Materials and Methods. Overall, this section is well-structured, and all protocols and techniques used are specified.
Response: We appreciate this thoughtful feedback.
- In section 2.4, Clinical Data Collection and Characterization, on line 162, the authors explain that the classification of SIRS is based on the article by Roy et al. It appears that in their objectives, they establish a classification into four groups (SIRS1, SIRS2, SIRS3, and SIRS4) following the methods applied in Roy et al.'s article. However, it is not specified whether they considered lactate levels and mucous membrane color, as concluded in that article, to increase sensitivity and specificity. Could the authors clarify this point and include this information in this section?
Response: Thank you for highlighting the need for clarification regarding SIRS scoring in this study. We only used the 4 central criteria (abnormal heart rate, respiratory rate, temperature, and peripheral white blood cell count) when determining SIRS score for the purposes of this study. We did not include lactate concentration and mucous membrane color as proposed for the “Severe SIRS” classification in Roy et al. (2017). We have added details to our Methods section to make this more clear.
- The results and discussion are clear, and the authors have been very transparent about the limitations. It would have been preferable not to include horses that were euthanized for economic reasons in the statistical analysis (at least in the outcome part of the study). However, this might not have changed the results, as this biomarker also did not allow differentiation between types of colic.
Response: Thank you for this comment. As mentioned in the Discussion section, to our knowledge, none of the horses in this study were euthanized purely for economic reasons. However, we recognize that finances are at least one of the factors that drive client decision making across veterinary medicine, including whether to pursue surgery or more intensive medical management in colic patients. Because all of the non-survivors in this study were euthanized (none died), we are unable to exclude the possibility that economic reasons contributed to the clients’ decisions for euthanasia.
Round 2
Reviewer 2 Report
Comments and Suggestions for Authors
Minor Comments:
- Abstract: Consider simplifying the sentence starting with "Identification of novel diagnostic and prognostic biomarkers..." for clarity. It could be broken into two sentences to enhance readability.
- Introduction, Line 35: The phrase "advising owners on prognosis, making decisions regarding..." could be streamlined to "advising owners on prognosis and making decisions regarding..."
- Methods, Line 105: The section could benefit from clearer subheadings or bullet points to delineate the different groups of horses being studied (strangulating, non-strangulating, inflammatory, etc.).
- Results, Line 185: Consider adding a table or figure summarizing the key statistical comparisons for easier reference, as the text is currently dense with numeric data.
- Discussion, Line 301: The sentence beginning "In this study, there was no significant difference..." could be rephrased to avoid redundancy with previous statements about cfDNA levels.
- References: Ensure that all references are in a consistent format, particularly regarding the use of italics for journal names and proper abbreviation.
- General Formatting: Review the document for consistent use of font, spacing, and alignment throughout tables and figures to improve overall presentation.
Author Response
Comment 1: Abstract: Consider simplifying the sentence starting with "Identification of novel diagnostic and prognostic biomarkers..." for clarity. It could be broken into two sentences to enhance readability.
Response 1: We have revised the specified sentence to improve readability.
Comment 2: Introduction, Line 35: The phrase "advising owners on prognosis, making decisions regarding..." could be streamlined to "advising owners on prognosis and making decisions regarding..."
Response 2: We appreciate the reviewer's feedback. However, we respectfully maintain that the suggested change would alter what we were trying to convey in this sentence (4 separate challenges); the commas are needed to separate the 4 items on the list. We have added a colon after "include" to help signify that a list was coming.
Comment 3: Methods, Line 105: The section could benefit from clearer subheadings or bullet points to delineate the different groups of horses being studied (strangulating, non-strangulating, inflammatory, etc.).
Response 3: We appreciate the reviewer's suggestion. In our version of the word document of the manuscript, Line 105 is at the end of the introduction and it describes the three types of comparisons we plan to make, 1. lesion, 2. SIRS status, 3. survival to discharge. We use numbering as a literary tool to illustrate the three major comparisons. Subheadings or bullets would interrupt the flow of the paragraph structure of the introduction.
Comment 4: Results, Line 185: Consider adding a table or figure summarizing the key statistical comparisons for easier reference, as the text is currently dense with numeric data.
Response 4: We appreciate the reviewer's suggestion. Each paragraph of reported results is supported in an existing Table, Figure, Supplemental Table, or Supplemental Figure. We have bolded the text referring to the images and tables.
Comment 5: Discussion, Line 301: The sentence beginning "In this study, there was no significant difference..." could be rephrased to avoid redundancy with previous statements about cfDNA levels.
Response 5: We appreciate the reviewer's suggestion. This is the topic sentence for a discussion paragraph that goes on to compare and contrast our findings with other cfDNA research and is therefore important. However, we have revised several other occurrences of the phrase "in this study" to help with the repetitive use of this phrase. Specifically in lines 311, 328, 337, 380.
Comment 6: References: Ensure that all references are in a consistent format, particularly regarding the use of italics for journal names and proper abbreviation.
Response 6: We have reviewed the list of references, and they all appear to be formatted correctly (italicized journal names).
Comment 7: General Formatting: Review the document for consistent use of font, spacing, and alignment throughout tables and figures to improve overall presentation.
Response 7: We thank the reviewer for bringing the formatting error in Figure 2 to our attention. We have corrected the mis-aligned x-axis labels in Figure 2, and the rest of the document/figures appear consistent.